# A New Electric Field Mill Network to Estimate Temporal Variation of Simplified Charge Model in an Isolated Thundercloud

**DOI:** 10.3390/s22051884

**Published:** 2022-02-28

**Authors:** Kozo Yamashita, Hironobu Fujisaka, Hiroyuki Iwasaki, Kakeru Kanno, Masashi Hayakawa

**Affiliations:** 1Faculty of Engineering, Ashikaga University, 286-1 Omae-cho, Ashikaga 326-8558, Japan; kannokkr@gmail.com; 2Fujisaka Technology Office, 1058-5 Yoshizawa-cho, Ōta 373-0019, Japan; uii19005@nifty.com; 3Cooperative Faculty of Education, Gunma University, 4-2-Aramaki-machi, Maebashi 371-8510, Japan; iwasaki@gunma-u.ac.jp; 4Advanced Wireless & Communications Research Center, University of Electro-Communications, 1-5-1 Chofugaoka, Chofu, Tokyo 182-8585, Japan; hayakawa@hi-seismo-em.jp

**Keywords:** electrostatic field, electric field mill, thundercloud

## Abstract

The gross charge distribution in an electrified cloud has already been estimated by polarity distribution of the electrostatic field on the ground surface. While either a dipole or a tripole charge structure is commonly accepted, the increase–decrease and motion of each point charge in those models are both still unclear. This paper presents a new network of electric field mills for multipoint electrostatic measurement to evaluate the temporal variations of a simple cloud charge model with second-scale resolution. Details of our newly developed equipment are described, with an emphasis on its advantages. This network was deployed in the north Kanto area of Japan and operated during the summer season in 2020. In order to simplify the relationship between cloud charge positions and the horizontal distribution of the measured electrostatic field, an isolated thundercloud is focused on. As an initial analysis, a negative point charge model is applied to an isolated cloud observed on 27 August 2020. The quantity and height of the point charge were estimated as being approximately −20 C and 7 km, respectively. The calculated charge location is generally coincident with the C-band radar echo regions. Significant correspondence is demonstrated between the intensity distribution of the electrostatic fields measured at seven sites and that calculated with estimated point charge. This result indicates the possibility to determine the amounts and positions of cloud charges inside the dipole charge structure based on multipoint measurement of the electrostatic field.

## 1. Introduction

Electrification of thunderstorms is the origin of lightning discharges. It is essential for lightning protection to understand the electrical nature of a storm. The investigation of electrification inside a thundercloud can be divided into two main categories. One is research of the electrical structure of a thunderstorm, and the other is the quantitative estimation of charge amount inside a thunderstorm.

The electrical structure inside a thundercloud has been studied previously. In the early days, for thunderstorm researchers, the horizontal distribution of the electrostatic field on the ground was measured to estimate the electric structure inside a thundercloud [1,2]. In general, the vertical structure of the electric charge inside a cloud was considered to be a dipole or a tripole structure [3].

The heights of the charge regions inside a cloud were estimated using electrostatic measurements and radio observations. Based on a multipoint measurement of electric field change (ΔE) caused by a lightning discharge, the heights and amounts of charges neutralized by lightning discharges could be estimated [4,5,6]. A three-dimensional (3-D) lightning location system based on radio observation allows us to map the radiation sources of lightning channels, which represents the heights of charge regions in a cloud [7,8,9].

The vertical distribution of the electrostatic field inside a thundercloud has also been measured by in situ balloon electrostatic measurements [10,11]. Balloon observation makes it possible to estimate a snapshot of the vertical structure of the charge inside a cloud, which is more complicated than a dipole or tripole charge model [12]. A combination of the 3-D mapping of lightning channels and in situ balloon electric field measurements shows the relationship between the 3-D propagation path of the lightning channel and the height of the major charge region [13]. Comparison between the vertical distribution of the radiation sources of the lightning channel and the 3-D structure of radar echoes also enables us to infer the height of the charge region inside the cloud [14].

While the charge structure of a thundercloud is estimated by recent advanced techniques, such as 3-D lightning mapping [7,8,9], there are few works that have estimated the amounts of charges in an assumed simple charge model. The amounts and heights of negative and positive charges in the dipole charge model are calculated by measuring the electrostatic field above a thunderstorm by airplane [15]. Simultaneous use of radio observation for 3-D lightning channels and multipoint electrostatic measurement on the ground had also been carried out to estimate the positions and amounts of charge centers, respectively, inside a thunderstorm [16].

Electrostatic measurement on the ground is a classical way to detect electrification inside a storm. Though the electrostatic field near, beneath, and inside a thundercloud has already been measured in previous studies, it is still difficult to understand the relationship between the temporal changes of the surface electrostatic field and the growth or declination of electrification in a cloud. Not only the growth of the charge inside a thundercloud but also the approach of a thundercloud to the sensor enhances the magnitude of the electrostatic field on the ground surface. The increase–decrease and motion of the charge center inside a thunderstorm are still unclear.

One of the goals in this study is to estimate the temporal variation of point charges in classical dipole or tripole charge structures with second-scale resolution. It is not realistic to determine the locations of the charge centers in a wide-spreading cloud. In this study, a single-cell thunderstorm, which was spatially isolated, was focused on to simplify the relationship between the horizontal distribution of the electrostatic field on the ground and the charge centers hypothesized inside a cloud. By assuming a simple charge structure in an isolated cloud, the temporal variations of the charge centers inside a thundercloud can be derived.

This paper presents a new network of electric field mills (EFMs) for multipoint electrostatic measurement, which were constructed in the north Kanto region, Japan. An original EFM system was designed and fabricated in this study, and we here emphasize the advantages of this system. This EFM network was operated during the summer season in 2020. As an initial result, the electrostatic waveforms observed at seven sites, which were associated with the isolated thundercloud observed on 27 August 2020, were analyzed. By assuming a negative point charge structure inside the cloud, the location and amount of idealized point charge was calculated with second-scale resolution. This result is a touchstone to evaluate the increase–decreases and motions of charges in the dipole charge structure by using horizontal distributions of the surface electrostatic field.

## 2. Observation

### 2.1. Sensor: Electric Field Mill (EFM)

#### 2.1.1. Sensing Module

As a sensor system for electrostatic measurement, an original electric field mill (EFM) was designed and fabricated in our laboratory. The assumed parameters are summarized in Table 1. There are some original points in signal processing and synchronization of the sensors, which was necessary to construct our observation network. Details of our design for signal processing are described in Section 2.1.2 and Section 2.1.3.

A picture of our sensor system, developed and deployed in the summer season of 2019, is shown in Figure 1a. A sensing module of the EFM was installed facing downward to eliminate the effect of charged raindrops. Modules for data recording and power supply were stored in the observation box of Figure 1a.

A sensing module consists of sensing plates, a rotating blade, and a signal amplified circuit. The shape and size of these plates were designed in our laboratory. The metal parts, such as the sensing plates and the rotating plate, were cut from duralumin plates using a desktop Computer Numerical Control (CNC) milling machine. The parts of the sensing module are shown in Figure 1b.

The rotating blade should be grounded to shield the sensing plates, and this blade is rotated with a motor. A brushless motor (McLennan Servo Supplies: BLDC48-12L-037) was used due to the low noise properties of these motors. A ball bearing (NTN: 696LLU) is mounted on the motor shaft and connects the rotating blade with the base plate, which is also grounded electrically. Carbon grease (K-CON: 0900-969-00160) is injected inside the ball bearing to keep the resistance small. The resistance between the rotating blade and the grounded base plate was measured as less than 100 Ohms. An identical method was adopted in the previous study [17].

Rotation speed is set as slow as possible to expand the life of the equipment. The rotation of the blade is monitored by the proximity sensor. The rotation signal from the proximity sensor is used for two purposes. One is to determine the direction of the electrostatic field on the ground; details for determination of the direction of the electrostatic field are described in Section 2.1.2. The other is to keep the rotating speed constant. The output signal from the sensing plate is an AC voltage waveform whose frequency is maintained at 10 Hz. Details are summarized in Section 2.1.3.

#### 2.1.2. Signal Processing and Data Recording

Both the magnitude and the direction of the electrostatic field on the ground surface is measured using our developed EFM system. A block diagram of the signal processing is summarized in Figure 2. The signal to determine the magnitude of the electrostatic field, which is described as the “Es signal” in Figure 2, is generated from the sensing plate. The sensing plates are opened and shielded periodically and yield a sinusoidal curve whose amplitude is coincident with the magnitude of the electrostatic field. This sinusoidal curve, named the “Es signal” in Figure 2, is amplified by an analog circuit inside the sensing module. The gain of the analog circuit is set as 40 dB. The radius of detection for the electrified thunderstorm from each sensor is evaluated experimentally as approximately 30 km.

The amplified electrostatic signal is fed to the data recording system and converted to digital signals by an analog-digital (AD) converter (MINGYUANDINGYE: ADS1115) with a resolution of 16 bits. The timing of the sampling is determined using a pulse per second (PPS) signal from a GPS module (YIC: GT-902PMGG). The PPS signal is read by a microcomputer (STM32 L432KC) as a digital signal and used to correct the internal clock of the microcomputer. The sampling period is set as 10 ms.

A proximity sensor is used to monitor the speed and timing of the blade rotation. These analog signals are also transferred from the sensing module to the microcomputer (STM32 L432KC). The period of rotation signal from the proximity sensor is checked by the microcomputer, and the “rotation signal” in Figure 2 is constantly controlled at 100 ms. Details are summarized in Section 2.1.3. The timings and amplitudes of the maximum and the minimum point of the “Es signal”, described within 100 ms, are extracted and saved as parameters on an SD card. Ten samples are recorded within 1 s. In this paper, an averaged amplitude during 1 s is used as the magnitude of the electrostatic field at each station.

#### 2.1.3. Synchronization of EFM Systems

One of the most important issues when constructing and operating a network for multipoint electrostatic measurement is time synchronization of all the sensors. The rotation speed of our EFM is relatively slow. A variance of timings for the opening and shielding of the sensing plate would affect the magnitude of the electrostatic field. If the rotation at each station cloud is not synchronized, a random error could not be ignored when an electric field change (ΔE), which is a rapid change of the electrostatic field due to lightning discharge, is measured. In this study, not only the timing of AD conversion in the data recording system but also the rotation of the grounded blade is synchronized.

The synchronization of AD conversion has already been described in Section 2.1.2. For synchronization of the rotation of the grounded blade at each station, both the rotation signal from the proximity sensor and the PPS signal by the GPS module are used. Initial rise timing of the rotation signal is controlled to fit that of the PPS signal based on PI control. Rotation speed is also controlled in such a way that the period of the rotation signal is constant at 100 milliseconds.

The temporal coincidence between the rotation of the grounded plate and the GPS signal is monitored by the microcomputer (STM32 L432KC), as described in Figure 2. Figure 3 shows an example of a rotation signal from the proximity sensor and a PPS signal from the GPS module. Initial rise timing of the rotation signal and that of the PPS signal from the GPS module are generally coincident. The inconsistency of the rotation of the grounded plate at each station is evaluated as less than 5 ms. The period of rotation signal is 100 ms, and so the synchronization error of the rotation of the grounded plate is estimated at approximately 5%.

### 2.2. Calibration

In our EFM network, the sensor was installed on the ground or on a rooftop. The sensitivity of our EFM system is affected by deformation of the electric field line due to the shape of the landform or the buildings on which the EFM systems were installed. The magnitudes of electrostatic fields measured at various altitudes should be converted to that at the flat ground level for the multipoint electrostatic measurement. The methodology of such calibration for the sensors is summarized in this section.

The calibration of the EFM network in this study was carried out based on the simultaneous measurement of the homogeneous downward electric field at each station during fair-weather conditions, in which no cloud could be identified. During the summer season of Japan, while thunderstorm activities are enhanced, there are few days with fair-weather conditions. It is difficult to measure the fair-weather electric field in parallel with thunderstorm observation. The fair-weather electric field could be measured relatively easily during fall season. Therefore, we continued to measure the fair-weather electric field from August to December in 2020.

In order to measure the fair-weather electrostatic field at ground level (E_FG_ [V/m]), an EFM system was temporally installed on the ground surface. A picture of the EFM system on the ground, which was covered by a grounded metal plate, is shown in Figure 4. Output, which is consistent with E_FG_ [V/m] at each site, was converted to the strength of E_FG_ [V/m] measured by the sensor of Figure 4.

To identify fair-weather conditions, both the capture of sky images by a camera system and visual confirmation was carried out. As a camera system, a compact small single-board computer (Raspberry Pi 3 Model B+) and a camera module for this computer were used. Sky images above sensors were captured every minute. There were conditions in which no clouds could be recognized by our camera system, but a light cloud could be confirmed by visual confirmation. Data for calibration in this paper was confirmed by both a camera system and by visual confirmation.

Examples of waveforms for calibration are shown in Figure 5. The bottom panel indicates the waveform of E_FG_ [V/m] on 16 November 2020. The top panel shows the waveforms of output voltages that were measured at five sites on the same date. A period of fair-weather condition is recognized during the period 11:00–11:30 (LT) on 16 November 2020 by monitoring the sky images of the camera system. Visual confirmation was also carried out at the calibration site, as described by a blue circle in Figure 6. Thin low cloud, which had not been recognized by the camera system, was identified by visual confirmation around the period 11:00–11:30 (LT). In this case, waveforms during the period 11:10–11:20 (LT) are used as the fair-weather electric field for the calibration of sensors. In this paper, we collected the waveforms of fair-weather condition from August to December in 2020 and correct the sensitivity of the sensors at each station.

It is difficult to define fair-weather conditions clearly. The criteria of conditions to measure the fair-weather electric field has been discussed in preceding studies [18,19,20]. Furthermore, we also verified the fluctuation of the surface electrostatic field, although a fair-weather condition could be recognized by using both camera system and visual confirmation. One of the reasons for this fluctuation is wind, which affects the spatial distribution of the space charge locally. We cannot eliminate this wind effect. This ambiguity would cause an error of calibration and affect the inaccuracy of magnitudes of the electrostatic field at each site. The errors of the calibrated electrostatic fields at all stations were estimated as less than 10%.

### 2.3. Sensor Network

Sensors were distributed in the south region of Tochigi and Gunma prefectures in Japan, which is a dynamic zone for thunderstorm activity [21,22]. In this study, electrified clouds that occurred on 27 August 2020 were focused on as spatially isolated thunderclouds. The positions of the EFM systems that were running during the approach of the thundercloud are described as red circles in Figure 6. The location for the measurement of the surface electrostatic field at ground level E_FG_ [V/m] is also described as a blue circle in Figure 6.

The black dots in Figure 6 indicate lightning distributions associated with the thundercloud analyzed in this paper. Lightning data was obtained from the Japanese Lightning Detection Network (JLDN). In this dataset, the accuracy of geolocation is 300 m, and the detection efficiency of cloud-to-ground (CG) lightning discharge is estimated as being over 90%.

## 3. Data Analysis

In this paper, temporal changes of the idealized point charge inside a thundercloud are estimated based on multipoint electrostatic observation. As an initial analysis, we presume and estimate a negative point charge structure inside an isolated thundercloud because the number of stations is not yet sufficient. The surface electrostatic field $E_{Ci}\left(x_{i},y_{i}\right)\;$ at the *i*-th station due to the assumed point charge is described as follows:(1)$$E_{Ci}\left(x_{i},y_{i}\right)=\frac{Q\cdot z_{C}}{2\pi \varepsilon_{0}\cdot {\left\{{\left(x_{C}-x_{i}\right)}^{2}+{\left(y_{C}-y_{i}\right)}^{2}+z_{C}{}^{2}\right\}}^{1.5}}\left[V/m\right]$$

The coordinate system of the point charge model is described in Figure 7. There are four unknown parameters that represents position ($x_{C}$, $y_{C}$, $z_{C}$) and amount (*Q*) of the assumed point charge. The position of the *i*-th station is represented by ($x_{i}$, $y_{i}$). More than four stations are needed for electrostatic measurement to determine the four variables, $x_{C}$, $y_{C}$, $z_{C}$, and *Q*.

In the previous studies [4,5,6], the chi-squared test was used to calculate the positions and quantities of the charges neutralized by a lightning discharge. An approach using the analysis of inverse problem is also used to derive the electric charge in the thundercloud [16]. In this paper, the chi-squared test is used to determine the location and intensity of the point charge as an initial analysis.

The theoretically calculated electrostatic fields were compared with those observed by our EFM network. We set the grids as described in Table 2. The resolutions of longitude and latitude were configured as 0.005 degrees. The values of the surface electrostatic field at each site were calculated by assuming the point charge, which has *Q* Coulomb positioned at each grid.

The comparison between the values of the observed electrostatic field at each station and that computed using the surface electrostatic field is as follows. As criteria to determine the position and amount of point charge, $\chi_{\nu }{}^{2}$ is calculated by:(2)$$\chi_{\nu }{}^{2}=\frac{1}{\nu }\cdot \overset{n}{\underset{i=1}{\sum }}\frac{{\left(E_{Oi}-E_{Ci}\right)}^{2}}{\sigma_{i}{}^{2}}$$
where *E_Oi_* [V/m] is the magnitude of the surface electrostatic field measured at the *i*-th station, *E_Ci_* [V/m] is the calculated electrostatic field based on the point charge model, the value of *σ_i_* ^2^ represents the estimated variance of the electrostatic measurement at the *i*-th station, *n* is the total number of sensors, and *σ_i_* was evaluated as 10% of the calibrated amplitude of the electrostatic field at each station. This value is based on the error evaluation of calibration, as described in Section 2.2. The value of *ν* is the number of degrees of freedom, which is the surplus number of observation sites. The value of $\chi_{\nu }{}^{2}$ is used to analyze the thundercloud on 27 August 2020, which was monitored at seven sites. The number of unknowns is 4 and that of measurements is 7, so the number of *ν* is calculated as 3.

If the horizontal distribution of the surface electrostatic field calculated based on the assumption of the point charge model is correct, the value of $\chi_{\nu }{}^{2}$ becomes approximately or less than 1.0 [23]. We determined the position and quantity of the point charge, which provided a minimum value of $\chi_{\nu }{}^{2}$ as the best result to fit the observed and calculated values.

The limitation has to be specified to select the charge position. In this paper, we only extracted the grids whose heights were over 6 km to geolocate the point charge in the cloud. If this restriction is not set, the solution of charge position becomes unstable, and the grids whose heights are approximately 4 km are also selected. The propriety of this limitation should be checked by further analyzing isolated thunderclouds.

## 4. Results

### 4.1. An Isolated Thundercloud on 27 August 2020

The waveforms of the surface electrostatic fields obtained at seven sites during the period 0:00–3:00 (LT) on 27 August 2020 are shown in the top panel of Figure 8. During this period, an isolated thundercloud occurred and approached our EFM network. The number of sensors running during the approach of the thundercloud was seven. Omae, Fukui, Honjo, Kawasaki, Yoshizawa, Narushima, and Hagari are the names of the towns in which sensors were running when the thundercloud occurred.

The waveforms observed at the seven stations were found to be generally in agreement. The amplitudes of the electrostatic fields observed at the seven sites reached maximum at approximately 1:40 (LT). To infer the phase of this thundercloud activity, the lightning data obtained by JLDN and radar echoes were examined. The frequency of the lightning discharges associated with this thundercloud every 10 min is also summarized in the bottom panel of Figure 8. The lightning occurrence in Figure 8 includes CGs and ICs. The spatial distribution of the lightning discharges is shown in Figure 6.

While the peak of lightning occurrence was 1:00–1:10 (LT), the maximum amplitudes of the electrostatic fields were observed at approximately 1:40 (LT). There is a difference between the timing of the maximum amplitudes of the surface electrostatic fields and that of the peak of lightning occurrence. This result suggests that the mature stage of this thundercloud would be before 1:10 (LT). The maximum amplitudes of the electrostatic fields obtained by our EFM network would be caused by the approach of this thundercloud, and the thundercloud was in the decline phase at approximately 1:40 (LT).

The distribution of the rain intensity of this isolated thundercloud at 2 km height in the vicinity of our EFM network is also shown in Figure 9a,b. In Figure 9, the locations of the sensors are indicated as red dots. The black and red lines describe the shapes of the echo regions in every 1 km and 5 km, respectively. The echo region by C-band radar at 1:40 (LT) is shown in Figure 9a, which suggests that the shape of the echo region is spatially isolated.

After 1:40 (LT), the echo region of this isolated thundercloud became small and then disappeared. Figure 9b demonstrates that the echo region at 2:10 (LT) is obviously dissipated. This result also supports the view that this thundercloud would be in the dissipating stage at approximately 1:40 (LT).

The significant coincidence of the waveforms observed at the seven sites in Figure 8 is caused by the isolated thundercloud that approached the EFM network. This result could be taken as an indication that the temporal variations of the surface electrostatic fields at the seven sites were caused by the same source. In Figure 8, the negative point charge structure is imaged inside this isolated thundercloud as a source of the electrostatic field.

One of the doubts concerning the observation of this isolated thundercloud is a drastic variation of the electrostatic field at approximately 1:10 (LT) at Hagari, where there is a croft. This fluctuation could be caused by local phenomena, such as the approach of a neighboring resident, but we could not specify the details. We analyzed the electrostatic waveforms to estimate the height and amount of charge inside the thundercloud after 1:30 (LT). Additionally, this thundercloud collapsed, and its shape became complex after 1:50 (LT), which indicates the declining phase of the thundercloud.

Figure 10a,b shows an example to estimate the position and amount of assumed point charge inside the thundercloud at 1:40 (LT). The blue square in Figure 10a demonstrates the longitude and latitude of the point charge model, calculated with the amplitudes of the electrostatic fields at 1:40:00 (LT). The estimated position of assumed point charge is generally consistent with the radar echo region shown in Figure 10a.

Figure 10b represents the result of fitting between the observed and calculated amplitudes at each station. The horizontal axis shows the distance from the position of charge inside the thundercloud to the location of the sensors on the ground. The solid line in Figure 10b shows the horizontal distribution of the surface electrostatic field computed by Formula (1), with the amount and position estimated by the approach in Section 4. In Figure 10b, the coincidence between the observed horizontal distribution of the electrostatic fields and the calculated one is also demonstrated. This result supports the validness of our approach based on the assumption of the point charge model.

Figure 11 shows the temporal changes in the quantity and height of the point charge in the cloud over a period lasting 7 min. We selected the position and amount of assumed point charge, making the value of $\chi_{\nu }{}^{2}$ be the minimum. The discrete change in Figure 11 could be caused by the resolution of the geolocation of charge in the cloud. Temporal variations of height and amount of point charge in the isolated thundercloud during the period 14:39:00–14:45:59 (LT) are represented in Figure 11a,b. The calculated values of $\chi_{\nu }{}^{2}$ are also demonstrated in Figure 11c. The electrostatic waveforms at the seven sites are shown in Figure 11d.

Before 1:40 (LT), the value of $\chi_{\nu }{}^{2}$ is relatively large. This result indicates that it is not reasonable to apply the point charge model inside the thundercloud before 1:40 (LT). The reason for the increase of $\chi_{\nu }{}^{2}$ is discussed in Section 5. After 1:45 (LT), the signal-to-noise ratio becomes small due to the reduced electrostatic fields. In this paper, the period during 1:40–1:45 (LT) is considered to be the temporal variation of the assumed point charge model.

One of the notable points is the temporal variation of amount and height of the negative point charge model in Figure 11a,b. The charge amount decreases from −20 C before 1:43 (LT) to −10 C after 1:45 (LT), while the charge height remains constant at approximately 7 km. This tendency would be coincident with the declination of this thundercloud. This result demonstrates that our analysis could separate temporal variation for the point charge model from the oncoming or fading of a thundercloud.

### 4.2. An Isolated Thundercloud on 15 August 2020

A few isolated thunderclouds were observed during the summer season in 2020. The electrostatic waveforms associated with an isolated thundercloud observed on 15 August 2020 are shown in Figure 12a. The pulses in these waveforms corresponded to the electric field changes caused by lightning discharges. The occurrence of lightning discharges indicates that this thundercloud would not be in the dissipating stage during the period 19:40–20:30 (LT).

Figure 12b shows radar echo at 20:00 (LT) on 15 August 2020. The colored contour presents an echo region at 2 km height. The black and red lines represent the echo-top height in every 1 km and 5 km, respectively. It suggests that this thundercloud could be identified as an isolated thundercloud. Figure 12b also demonstrates that this thundercloud evolved perpendicularly. This result also supports the idea that this thundercloud would be in the initial or mature stage at approximately 20:00 (LT).

The charge amount and height of the thundercloud shown in Figure 12b could not be estimated by assuming a monopole charge structure. The reason for this result is discussed in Section 5.

## 5. Discussion

In this paper, two isolated thunderclouds are analyzed using a monopole charge model. While one thundercloud could be analyzed by assuming a monopole charge model, this could not be applied to the other. A point charge model is too simple, while this model makes it possible to simplify the calculation of charge center in a thundercloud. In previous studies, it was concluded that the actual charge structure is more complex. In this section, we discuss the condition that allows us to apply this point charge model to an isolated cloud. Our observation indicates that the phase of the thundercloud activity would be one of the most considerable parameters for the validness of the monopole charge model.

An example that could not be analyzed based on the assumption of the monopole charge model is demonstrated in Figure 12a,b. The electrostatic waveforms on 15 August 2020, observed at seven stations, are shown in Figure 12a. The pulses in these waveforms corresponded to electric field changes caused by lightning discharges. The occurrence of lightning discharges indicates that thundercloud activity would be in the initial or mature stage.

The radar echo at 20:00 (LT) on 15 August 2020 is shown in Figure 12b. The colored contour presents an echo region at 2 km height. The black and red lines represent the echo-top height in every 1 km and 5 km, respectively. The radar echo data suggests that this thundercloud, in the initial or mature stage, would have evolved perpendicularly.

The charge structure can be imaged based on the electrostatic waveforms during the period 19:40–19:50 (LT). In this period, the electrostatic field at Akouda station (No. 7), which was over 20 km distant from the thundercloud, attenuated strongly, and the magnitude of the electrostatic waveform remained at zero. The magnitudes of the electrostatic fields drastically changed at Yoshizawa (No. 1) and Omae (No. 2), which were located beneath the isolated thundercloud. The polarities at these stations became positive, which indicates a downward electrostatic field, so that this variation would be caused by the lower positive charge in the cloud. In contrast, the amplitudes of the electrostatic fields at Ohta (No. 3), Honjo (No. 4), and Fukui (No. 5) became negative, which was coincident with upward electrostatic fields. This means that the effect of the middle negative charge region was dominant in these areas. The electrostatic field at Kawasaki (No. 6), which was over 15 km distant from the cloud, became positive. This result indicates that the electrostatic field caused by upper positive charge was dominant at Kawasaki (No. 6). The charge structure could be inferred as a tripole charge model, which consists of lower positive charge, middle negative charge, and upper positive charge.

The difference of polarities at each site prevents us from applying the assumption of the point charge model in the calculation of the quantity and height of a charge center inside the cloud. If the point charge model is hypothesized, the polarity of the electrostatic field at each station should be the same. The point charge model cannot be used for this case.

The condition in which the polarities of the electrostatic fields in our EFM network are identifiable should also be discussed. In Figure 13a, the charge structure inside the isolated thundercloud at 1:40 (LT) on 27 August 2020 is imaged by assuming a dipole charge model inside the thundercloud. The vertical structure of the thundercloud could be imaged as that of an anvil cloud, in which there exists the vertical shear of negative and positive charge regions in the dipole model.

In Figure 13b, the surface electrostatic fields are calculated as a function of horizontal distance from the negative charge region by the assumption that the vertical shear of the negative and positive charge regions in the dipole model exists. The amounts of charges are determined based on a previous study [15]. If the vertical shear of charges does not exist (dR = 0 km), the horizontal distribution of the electrostatic field (E(x)) is calculated as a black line in Figure 13b. In the range of x > 6 km, the polarity of the electrostatic field becomes positive, which means there is a downward vertical electrostatic field caused by an upper positive charge region. This result is not coincident with that of our observation on 27 August 2020.

We also computed the horizontal distributions of E(x) when the vertical shear exists. Polarity becomes negative constantly, as described by the red, green, and blue lines in Figure 13b. We conclude that the shear between the positive and negative charge regions, which occurred in the dissipating stage of thundercloud activity, is a necessary condition for adopting the monopole charge model.

## 6. Conclusions

This paper presents a new network of EFMs for multipoint electrostatic measurement to estimate the temporal variation of the simplified charge model inside a thundercloud with second-scale resolution. In this study, EFM systems were newly designed and fabricated. The details of our system are described, with an emphasis of its advantages, in Section 2.1. The EFMs were distributed in the north Kanto lightning-prone region, Japan. For the calibration of each sensor, a fair-weather electrostatic field was simultaneously measured as a sources of the surface electric field. Details are summarized in Section 2.2.

To assume a simple charge structure inside an electrified cloud, only an isolated thundercloud was focused on. As a trial observation, the electrostatic waveforms associated with an isolated thundercloud observed on 27 August 2020 were measured successfully at seven sites. By using those waveforms, the height and amount of charge center in the cloud were calculated based on the assumption of a negative point charge model inside the isolated thundercloud. This test is a touchstone to apply the dipole or tripole charge model to an isolated thundercloud.

The estimated position of the negative point charge assumed inside the isolated thundercloud is consistent with the echo regions obtained by the C-band radar network, as shown in Figure 10a. A significant correspondence between the intensity distribution of the electrostatic fields measured at the seven sites and that calculated with estimated point charge is also demonstrated in Figure 10b. These results support the validness of the negative point charge model to interpret the relationship between the horizontal distribution of an observed electrostatic field and a simplified charge structure hypothesized inside a thundercloud.

The temporal variation of the negative point charge is shown in Figure 11. In Figure 11a, the charge amount varied from −20 C to −10 C. This tendency would be coincident with the declination of thundercloud activity. We conclude that the temporal variation of the charge amount and height in the negative point charge structure would mean the decline of charge inside a cloud.

A necessary restriction to apply to the point charge model is discussed in Section 5. During the summer season in 2020, a few isolated thunderclouds were observed. One of the considerable restrictions to the point charge model is the phase of thundercloud activity. The period of isolated thundercloud shown in Figure 9 is considered to be the dissipating stage. The upper positive charge region and the lower negative one inside a dissipating thundercloud would be affected by vertical shear, as described in Figure 13a. The spatial bias of the charge regions would be a restriction to apply to the point charge structure inside a cloud.

In this paper, there is one clear advance for thundercloud observations based on electrostatic measurement. We only calculated the growth and decline of charge inside a cloud with second-scale resolution, although a too-simple charge model is presumed. In previous studies, the magnitude of the electrostatic field on the ground was affected by both the approach and the fading of the thundercloud and the growth/decline of the charge region inside a cloud, as measured by an EFM network. While only the point charge model is used in this paper, due to the inadequate number of deployed EFMs, these primary results indicate the possibility that the dipole charge model could be applied to data from multipoint electrostatic measurements.

## Figures and Tables

**Figure 1 sensors-22-01884-f001:**
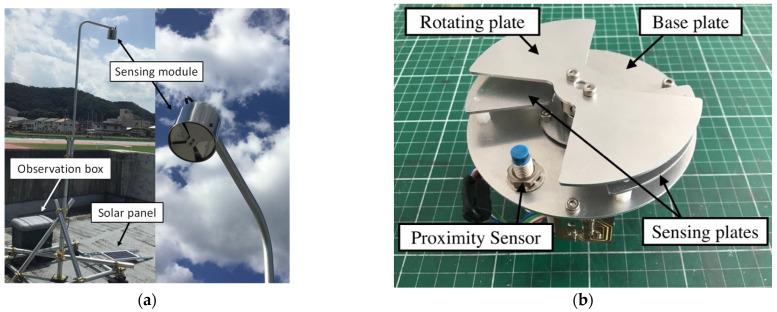
(**a**) Overview of the EFM system developed in this study (left) and picture of the sensing module (right). This system was installed at Honjo station. (**b**) Picture of sensing module, which consists of a base plate, sensing plates, a rotating plate, and a proximity sensor.

**Figure 2 sensors-22-01884-f002:**
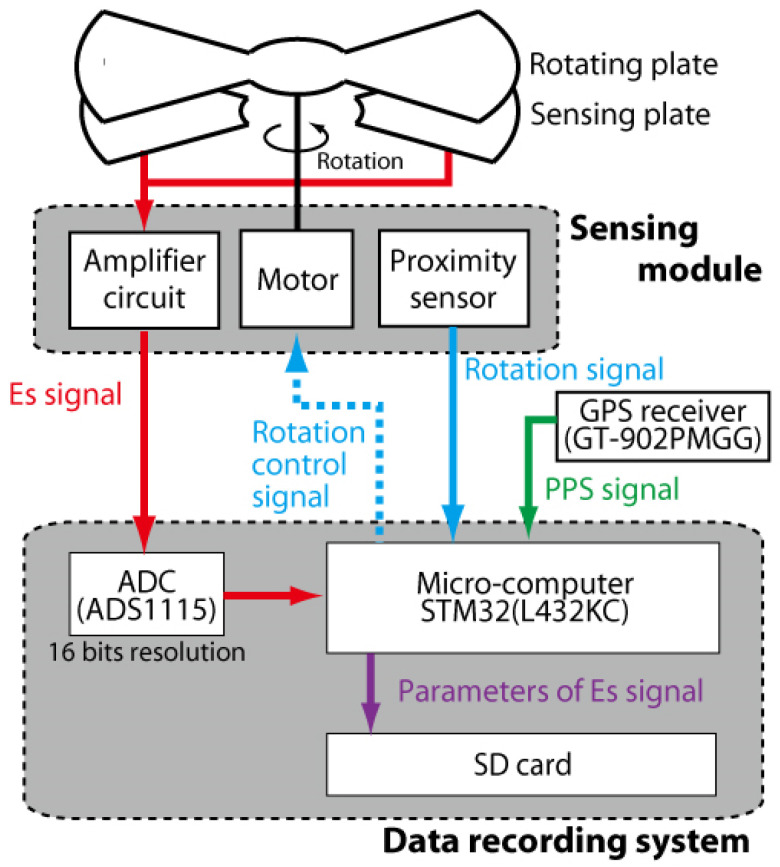
Block diagram of signal processing in our developed EFM.

**Figure 3 sensors-22-01884-f003:**
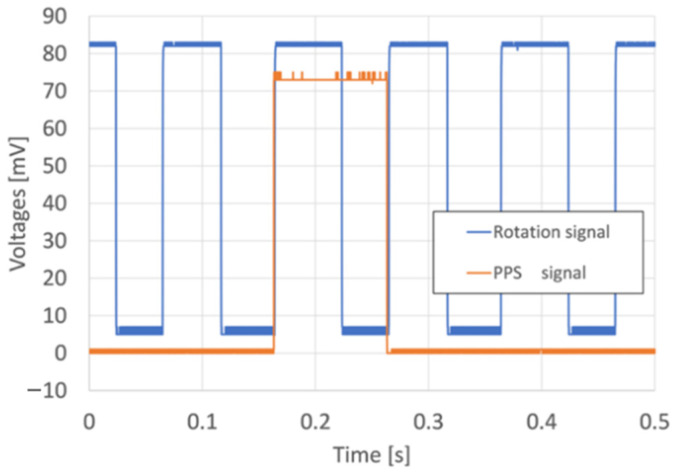
Example of a rotation signal from a proximity sensor (blue line) and PPS signal from GPS module (orange line).

**Figure 4 sensors-22-01884-f004:**
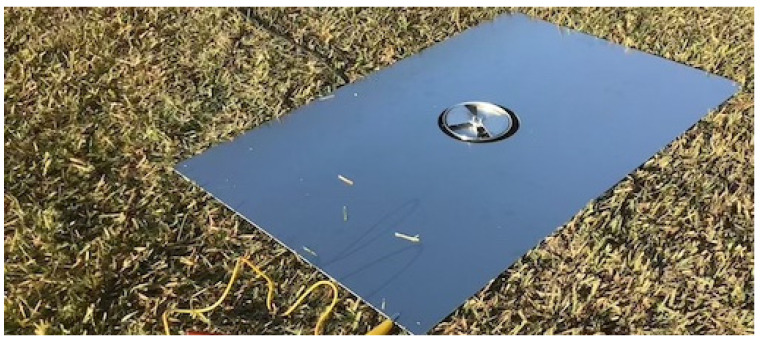
Picture of EFM system temporally installed to measure the fair-weather electric field at ground level.

**Figure 5 sensors-22-01884-f005:**
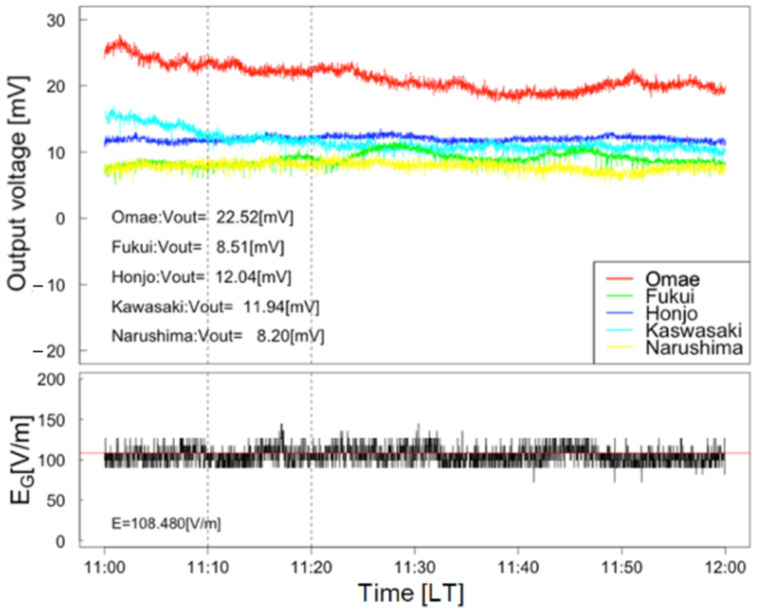
Top panel: Output voltages during fair weather obtained by sensors deployed in Ashikaga city during the period 11:00–12:00 (LT) on 16 November 2020. Bottom panel: Waveforms of fair-weather surface electrostatic field during the same period as top panel.

**Figure 6 sensors-22-01884-f006:**
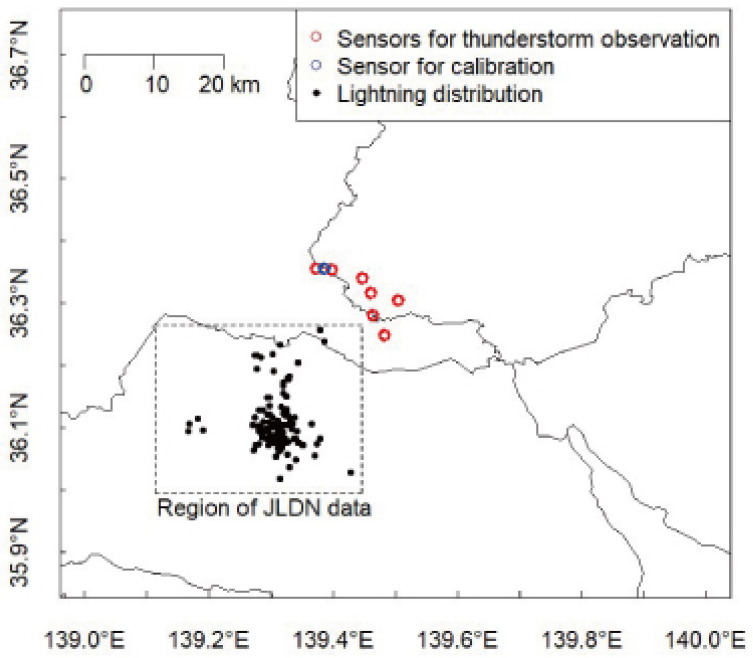
Distribution of the EFM systems running during oncoming thunderclouds on 27 August 2020. Blue circles are the location of a station for calibration. Black dots indicate the spatial distribution of lightning discharges obtained by the Japanese Lightning Detection Network (JLDN) during 0:40–1:40 (LT) on 27 August 2020.

**Figure 7 sensors-22-01884-f007:**
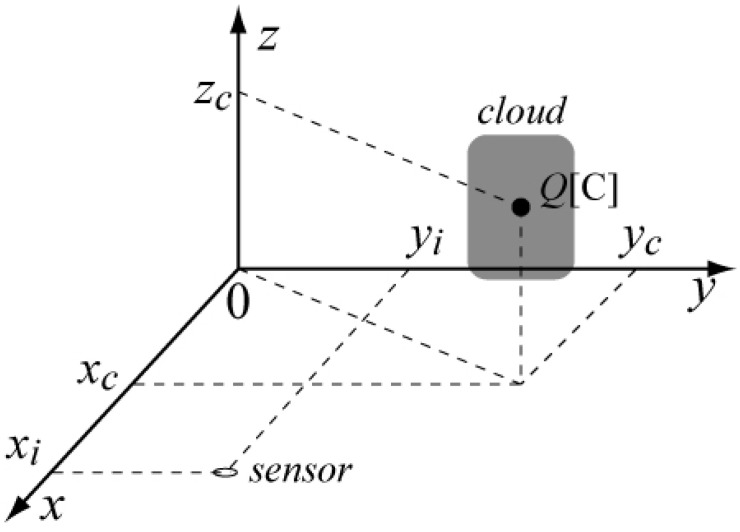
Coordinate system of the point charge model inside an isolated thundercloud.

**Figure 8 sensors-22-01884-f008:**
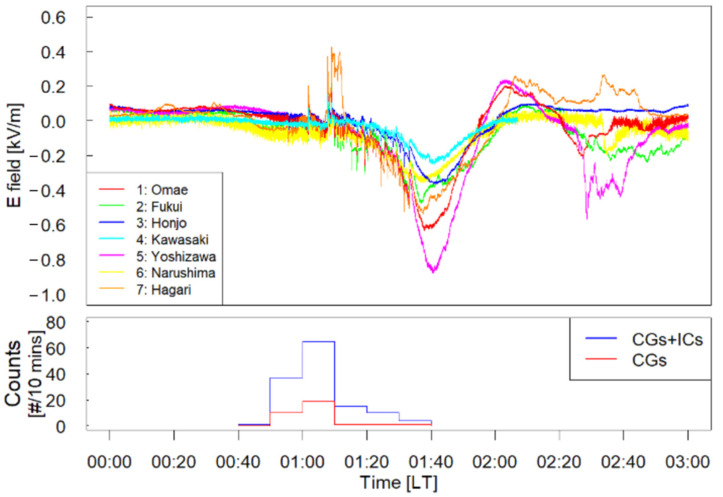
Top panel: Electrostatic waveforms near an isolated thundercloud during the period 0:00–3:00 (LT) on 27 August 2020. Bottom panel: Lightning occurrence obtained by JLDN during the period 0:40–1:40 (LT) (Adopted from [24]).

**Figure 9 sensors-22-01884-f009:**
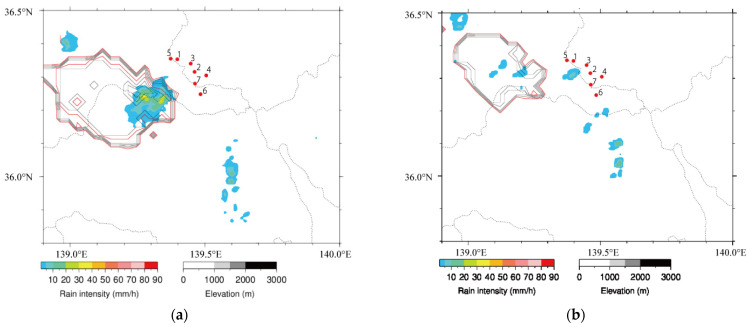
Distribution of rain intensity at 2 km height estimated by C-band radar at 1:40 (LT) (Panel (**a**)) and 2:10 (LT) (Panel (**b**)) on 27 August 2020. Black and red lines represent echo-top height in every 1 km and 5 km, respectively.

**Figure 10 sensors-22-01884-f010:**
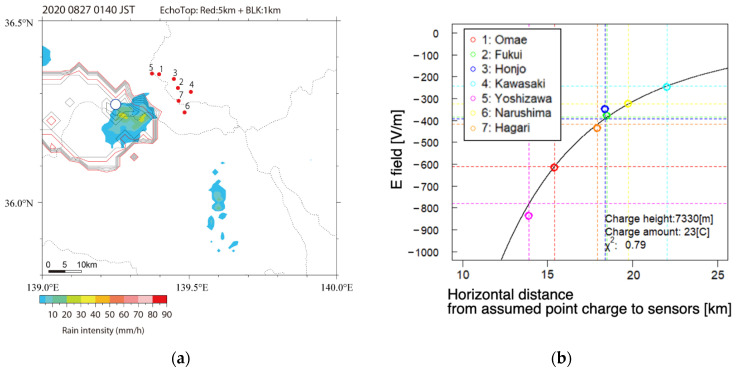
(**a**) Distribution of rain intensity at 2 km height, estimated by C-band radar at 1:40 (LT) on 27 August 2020. The blue circle indicates the estimated position of the assumed point charge in the cloud at 1:40:00 (LT). (**b**) The colored circles are the electrostatic field on the ground at each site. Horizontal axis shows the horizontal distance from the estimated position of charge to the sensors. The solid black line shows the result of fitting between the amplitudes of the electrostatic fields on the ground observed at seven stations and the calculated electrostatic field with the quantity and height of point charge determined by our method. (Adopted from [24]).

**Figure 11 sensors-22-01884-f011:**
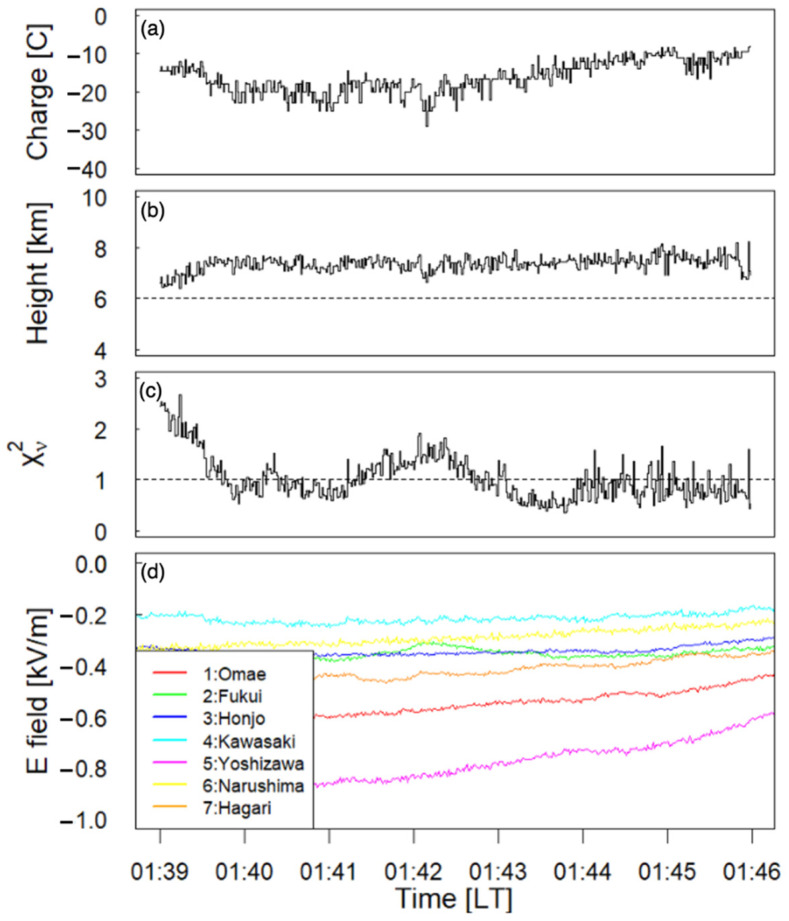
Panel (**a**) and (**b**): Temporal variations of the quantity Q (C) and height H (km), respectively, of the point charge model during the period 1:39–1:46 (LT) on 27 August 2020. (**c**) Value of $\chi_{\nu }{}^{2}$ during the same period of other panels. (**d**) Electrostatic waveforms obtained by our EFM network.

**Figure 12 sensors-22-01884-f012:**
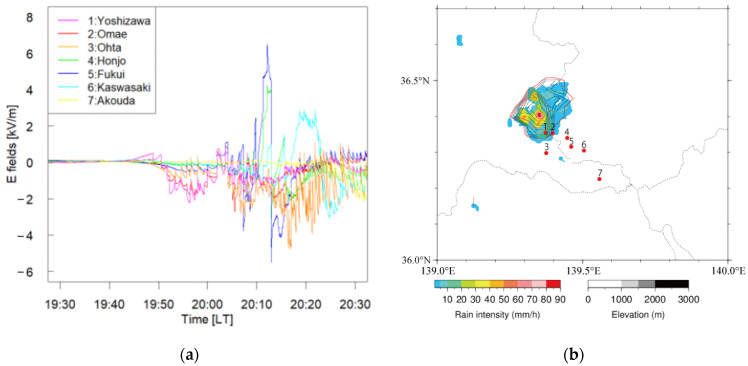
(**a**) Surface electrostatic waveforms beneath and near an isolated thundercloud during the period 19:30–20:30 (LT) on 15 August 2020. (**b**) Distribution of rain intensity at 2 km height estimated by C-band radar at 20:00 (LT) on 15 August 2020. Black and red lines represent echo-top height in every 1 km and 5 km, respectively.

**Figure 13 sensors-22-01884-f013:**
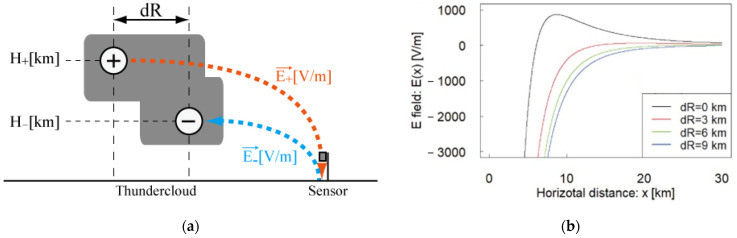
(**a**) Image for vertical structure of the isolated thundercloud observed on 27 August 2020, based on the echo structure of C-band radar. The value of “dR” is the horizontal distance of the positive and negative charges in the dipole charge structure. (**b**) Calculated horizontal distribution of surface electrostatic field on the assumption that “dR” between positive and negative could not be ignored. The horizontal axis is the horizontal distance from the charge inside a cloud to the sensor. The vertical axis is the electrostatic field on the ground.

**Table 1 sensors-22-01884-t001:** Assumed parameters of the developed EFM system.

Parameters	Details
Power-supply voltage	12 V
Power consumption	~1.6 W
Gain of amplifier	40 dB
A-D conversion	0–5000 mV (2450 mV bias), 16 bits resolution

**Table 2 sensors-22-01884-t002:** Setting of grids to calculate the surface electrostatic field at each station.

Parameters	Range	Intervals
Longitude	139.1 < L_N_ < 139.9 (deg)	0.005 (deg)
Latitude	36.0 < L_T_ < 36.7 (deg)	
Height of point charge	5 < H_Z_ < 20 (km)	10 m
Amount of point charge	−50 < Q < −1 (C)	1 (C)

## Data Availability

Not applicable.

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
