# Peer review of "A New Electric Field Mill Network to Estimate Temporal Variation of Simplified Charge Model in an Isolated Thundercloud"

_sensors, 2022, doi:10.3390/s22051884_

Round 1

Reviewer 1 Report

The mansucript by Yamashita et al. presents a new network of sensors to measure the electrostatic field of thunderclouds in Japan. The authors extensively describe the sensor and provide measurements of some cases. They employ the measurements to estimate temporal variations of charge amount and charge altitude inside the thundercloud by using a simplified charge model.

The topic is clearly in the scope of Sensors. The new network of sensors is clearly described, and the conclusions are supported by the results. However, I think the structure of the paper should be improved before publication.

I do not understand why the content of Sections 4.1 and 4.2 are separated. I think these two sections could be merged for simplicity. The same thundercloud is analyzed in both regions by using waveform measurements.

The authors present measurements of a new thunderstorm (August 15, 2020) in section 5, that is devoted to the discussion. I think that the measurements of this thundercloud could be included in Section 4 (results) instead of in the discussion section.

Minor comments:

  • I recommend consistence in the format of the dates: August 27th, 2020 or August 27, 2020.
  • Lines 57-64: I think including a description of Lightning Mapping Arrays (LMA) in this point could improve the introduction. LMA measurements can be used to estimate the charge structure of thunderclouds.
  • Lines 324-325: Please rephrase "Thundercloud would be in dissipating stage, so echo regions dissipated and separated." Do you means that the echo regions show the dissipation and separation of thunderclpuds, as expected?
  • Lines 337-338: Could the authors propose some local phenomena causing the measurements at Hagari?
  • Line 375: Please rephase "while charge height become a consist as about 7 km." Maybe something like: "while charge height remains constant at about 7 km."
  • Figure 1: Please add in which location is placed the sensor of the picture.
  • Figures 8, 11 y-axis: Fiels -> Field
  • Figures 9, 10, 12: Please add degrees N and degrees E as in Figure 6.

Author Response

We deeply appreciate your helpful comments. We agree your comments and revise our manuscript.

General comments:

The topic is clearly in the scope of Sensors. The new network of sensors is clearly described, and the conclusions are supported by the results. However, I think the structure of the paper should be improved before publication.

I do not understand why the content of Sections 4.1 and 4.2 are separated. I think these two sections could be merged for simplicity. The same thundercloud is analyzed in both regions by using waveform measurements.

The authors present measurements of a new thunderstorm (August 15, 2020) in section 5, that is devoted to the discussion. I think that the measurements of this thundercloud could be included in Section 4 (results) instead of in the discussion section.

We agree your comments “sections could be merged for simplicity”. We merge sections 4.1, 4.2 and 4.3 as “4.1 An isolated thundercloud on August 27th, 2020”.

Additionally, as you mentioned, observational results for a thunderstorm (August 15th, 2020) is included in section.4. We add electrostatic waveforms and radar echo data in section 4.2 as “4.2 An isolated thundercloud on August 15th, 2020” (lines 385-400).

Minor comments:

I recommend consistence in the format of the dates: August 27th, 2020 or August 27, 2020.

We revise our manuscript and use only “August 27th, 2020”.

Lines 57-64: I think including a description of Lightning Mapping Arrays (LMA) in this point could improve the introduction. LMA measurements can be used to estimate the charge structure of thunderclouds.

We revise lines 57-64. We add the sentence “While charge structure of a thundercloud is estimated by recent advanced techniques, such as 3-D lightning mapping [7-9], there are few works to estimate the amounts of charges in an assumed simple charge model.”.

Lines 324-325: Please rephrase "Thundercloud would be in dissipating stage, so echo regions dissipated and separated." Do you means that the echo regions show the dissipation and separation of thunderclpuds, as expected?

As you mentioned, this description is complicated. We revise lines 325-327.

Lines 337-338: Could the authors propose some local phenomena causing the measurements at Hagari?

Hagari station placed in a croft. There is a possibility for an oncoming of a neighborhood resident. We revised lines 336-339.

Line 375: Please rephase "while charge height become a consist as about 7 km." Maybe something like: "while charge height remains constant at about 7 km."

We deeply appreciate your point. We revise line 377.

Figure 1: Please add in which location is placed the sensor of the picture.

We thank your comments. In Figure.1, location of sensor is added. 

Figures 8, 11 y-axis: Fiels -> Field

Sorry for my mistakes. We revise y-axis of Figures 8 and 11. 

Figures 9, 10, 12: Please add degrees N and degrees E as in Figure 6.

We also add degrees N and degrees E in Figures 9, 10, 12.

Reviewer 2 Report

The authors constructed their own network of electric field sensors to measure the surface electric field change caused by the motion of electrified thunderstorms and the occurrence of lightning flashes. A lot of efforts have been paid to building this network, including some noval techniques that were not available tens of years ago. Certaintly this work is very excellent and it is suitable for publication on the journal of Sensors.

The prensentation of this work is also very good. I would only suggest the authors to find some sources of surface electric field measurements with caliabrated equipments in order to validate their results. 

Author Response

We deeply appreciate your essential comments.

The prensentation of this work is also very good. I would only suggest the authors to find some sources of surface electric field measurements with caliabrated equipments in order to validate their results.

We agree your comment. Actually, calibration of electrostatic sensor installed on the surface ground is one of the essential issues in this study. In this paper, we try to use fair-weather electrostatic field as a source of surface electric field measurements, as described in section 2.2. We add the sentence in line 470-472 to emphasize this point.